# Effects of Organizational Justice on Employee Satisfaction: Integrating the Exchange and the Value-Based Perspectives

Hyung-Woo Lee [1] and Dong-Young Rhee [2,*]

1   Department of Police and Public Administration, Hannam University, Daejeon 34430, Republic of Korea
2   Department of Politics and Public Administration, Hallym University, Chuncheon 24252, Republic of Korea
*   Correspondence: drhee@hallym.ac.kr

**Abstract:** Organizational justice is known to help promote organizational sustainability. The literature has explained the impact of organizational justice relying heavily on the social exchange perspective, the idea that employees are motivated to show favorable attitudes in return for the fair rewards that organization has provided. To the contrary, little attention has been given to the proposition that it affects employee's attitudes by increasing their intrinsic motivation. The latter has a greater implication of sustainable management since intrinsic motivation of its employees is the key to the sustained success of an organization. This approach can be called a value-based perspective. To fill that gap, this study examined the mediating effects of both the intrinsic and extrinsic motivation links between organizational justice and employee satisfaction. The analysis revealed that the indirect mediating effects of intrinsic motivation were greater than those of extrinsic motivation for procedural and interactional justice, while the mediating effect of extrinsic motivation was greater when it comes to distributive justice. In addition, the sum of the indirect effects of intrinsic motivation was comparable to that of extrinsic motivation. This result implies that the mediating effects of intrinsic motivation are as important as those of extrinsic motivation, confirming our prediction that organizational justice contributes to organizational sustainability via the path that has not been verified so far.

**Keywords:** organizational justice; intrinsic motivation; extrinsic motivation; federal employee viewpoint survey

## 1. Introduction

The issue of justice has always mattered for all types of organizations and, hence, has been extensively studied in almost all fields of the social sciences [1]. Particularly in the field of management studies, the concept of organizational justice has enjoyed extensive scholarly attention over the last few decades [2]. The recent research of justice in organizational contexts has demonstrated that promoting justice within organizations does have practical value as well: it has been shown to affect a variety of important attitudinal variables such as organizational citizenship behavior [3–5], organizational commitment [6–9], employee satisfaction [10–12] and a decrease in turnover intention [7,13].

Most scholars draw insights from social exchange theory to explain the mechanism by which organizational justice have positive effects on employees' attitudes [14–16]. Explaining the effect of organizational justice through the lenses of social exchange theory is predicated on the idea that employees in just and fair organizations are more committed to and satisfied with their jobs and organizations, mainly because organizations first showed commitment to the employees, and satisfied them by providing fair rewards. Translating this idea to the languages of motivation theory, it implies that organizational justice increases the job-related attitudes of employees through the mediating variable of extrinsic motivation. Although it is true that the social exchange perspective provides a sound and robust foundation for explaining the impact of organizational justice, there are more

reasons, beyond the pragmatic value of mutual exchange, why just and fair organizational practices have positive effects on job attitudes.

Although the exchange perspective does provide an insight to understanding the effect of organizational justice, the unbalanced view has led to a blindness to the additional and more profound value of organizational justice. The value-based perspective deserves equal attention for the following reasons. At first, justice has a value for its own sake. Justice is a noble virtue that we as a society must pursue regardless of whether it benefits or harms the interests of certain individuals and/or groups. In that sense, organizational justice signifies what values a given organization pursue, and the public awareness of these values contributes to the sustainability of the organization. More importantly, organizational justice not only contributes to achieving societal values, but also to the sustainability of organizations via the increased intrinsic motivation of its employees. Intrinsic motivation eventually increases organizational performance by reducing turnover [17,18], burnout [17,19] and deviant behaviors [20] which incur substantial managerial cost in the long run.

In fact, applying a social exchange perspective to organizational justice is relatively new, compared to the long history of the value-based perspective of justice [21]. To the extent that justice is primarily a philosophical value, the impact of organizational justice and the breach thereof will be manifested mainly through its effect on intrinsic motivation, rather than extrinsic motivation. This assertion is supported in part by the prior finding that value congruence is an important source of intrinsic motivation [1]. In other words, the breach of justice will lead to undermined value incongruence which subsequently results in the decrease in intrinsic motivation. However, it is striking to find that most empirical research on the topic is underpinned heavily on the social exchange perspective, and little research has examined the value-based perspective, namely, through the mediating path via intrinsic motivation.

In fact, there are several prior studies in this line of thinking, but the implications are somewhat limited. For example, Deschamps et al. [22] surveyed the employees in healthcare organizations in Canada to find that procedural and interactional justice showed significant influence on intrinsic (self-determined) motivation. Yet, this study fails to demonstrate the comprehensive picture of the motivational effect, Rather, it treated the justice variables as mediating variables. Hannam and Narayan [23] examined the effect of intrinsic motivation on organizational justice perception, but not the opposite causal relationship.

This study intends to fill this gap in the literature by examining the linkages of intrinsic motivation, as well as extrinsic motivation, mediating between organizational justice and job satisfaction. There exist two studies that examined the effect of organizational justice from the value-based perspective [24,25]. However, more investigation is still needed because Aryee et al. [24] tested the effect of overall justice, rather than the impacts of the three components of organizational justice separately. Moreover, Zapata-Phelan et al. [26] examined only the mediating effect of intrinsic motivation. To the contrary, in this study, the mediating effects of both extrinsic and intrinsic motivation will be tested in a single model that has three distinct justice components as exogenous variables. This study will make it possible to compare the relative sizes of each effect. This research will be meaningful both theoretically and practically because it will provide us with a more balanced perspective on the impact of organizational justice.

Moreover, if intrinsic motivation mediates the relationship between organizational justice and employee attitudes, it implies that the importance of organizational justice will be much greater than we have thought so far, because the literature suggests that the effect of intrinsic motivation on work attitudes is more significant than that of extrinsic motivation [27,28]. Notably, the extensive research on public service motivation (PSM) has proved that intrinsic factors are far more critical for public employees' motivation than extrinsic rewards [29]. Hence, this study will add an important theoretical basis for the imperative of promoting organizational justice. Although the degree of relevance of the

value-based perspective in managing public employees may vary depending on individual country's political, social and economic contexts [26], scholars have confirmed that it does matter for public employees' motivation in many countries [30,31].

To this end, in the remainder of this study, first, the hypotheses regarding not only the mediating linkage of extrinsic motivation but also intrinsic motivation, between organizational justice and job satisfaction, will be proposed. Second, using the 2019 federal employee survey, the structural equation analysis of the proposed causal relationship will be conducted. Third, the implications of the findings for scholars and practitioners will be discussed. The result of this study is expected to provide both theoretical and managerial implications.

## 2. Theory and Hypotheses

### 2.1. Social Exchange Theory and Extrinsic Motivation

Blau [32] provided a basis for understanding how a social exchange relationship can promote employees to behave in a way favorable to their organizations and supervisors. According to his formulation, social exchange relationships are sustained and developed by the expectation of reciprocity. In other words, organizations or supervisors initiate favorable treatment of their employee expecting that employees will reciprocate by exhibiting positive job attitudes. The examples of such favorable treatments cited most frequently in the literature are perceived organizational support (POS), and leader-member exchange (LMX). Engaging in social exchange relationships with organizations or supervisors creates a feeling of obligations on the part of individual employees, which plays a key role in eliciting positive attitudes from employees. Roch, Shannon, Martin, Swiderski, Agosta and Shanock [33] found that the sense of obligation to reciprocate is the primary driver behind a healthy exchange relationship between employees and their employers.

In this exchange-based relationship, employees exhibit a positive attitude from self-interest motives, in that they reciprocate the good deeds of organizations only when they perceive their exchange relationship with their organizations as valuable and worthy of further development. Rephrasing this through language of motivation theory [34], the proposition of social exchange theory implies that employees are extrinsically motivated to show positive work attitudes, expecting to reap the benefit of having exchange relationship with their organizations and supervisors.

This logic is commonly found in a number of research studies examining the effect of organizational justice from a social exchange theory perspective. For instance, Tekeleab et al. [35] found that a psychological contract breach, such as an unfulfilled expectation that organizations will reward employee's efforts, plays a key mediating role in predicting the effect of organizational justice on job satisfaction. Aryee, Budhwar and Chen [14] also found that trust and the employee's belief that organization will reciprocate their hard work (i.e., expectation related to social exchange), is a critical mediating factor of the process by which organizational justice results in positive employee's attitudes. Similarly, Rupp and Cropanzano [36] also found that organizational justice affects organizational citizenship behavior and organizational performance through the mediating variable of social exchange relationships. Masterson, Lewis, Goldman and Taylor [37] found that procedural justice perception affects organization-related outcomes such as satisfaction, commitment and intention to stay via the mediating variable of perceived organizational support. Masterson and his colleagues mapped their finding onto social exchange theory, explaining that organizational support is what employees receive from the organization, and commitment is what employees give the organization in return. Chen [15] also found that the perception of organizational justice increased the perception of social exchange and in turn suppressed their perceptions of a psychological contract breach. Based on these prior findings, this study also presents a hypothesis based on the same logic.

**Hypothesis 1.** *The effect of organizational justice on job satisfaction is mediated by extrinsic motivation.*

### 2.2. Value-Based Management Theory and Intrinsic Motivation

The value-based perspective is also relevant to examining the impacts that organizational justice has on work attitudes. A growing body of research is paying attention to the values as a motivating mechanism. The most salient example of this is the theory of public service motivation (PSM), which has recently gained considerable attention among public administration scholars. The key assumption of PSM theory is that public employees have a stronger public value orientation, which makes them motivated less by self-interest [38]. Scholars of PSM have found that employees with strong PSM will produce better work outcomes. Another example of the popularity of value-based perspective can be found in the leadership literature as well. The concept of transformational leadership has drawn the interest of scholars from various fields for the last thirty years [28]. Whereas transactional leadership was described as a more traditional model of leadership, transformational was initially proposed as a relatively new leadership style. Whereas transformational leadership relies heavily on the self-interest pursuits of employees, transformational leadership relies on the value pursuit of them. The term itself came from the underlying motivation mechanism of 'transforming' the values and beliefs of subordinates [39]. Similarly, it is also notable that Bass and Steidlmeier [40] identified the sound moral values of leaders as the key element of authentic transformational leadership that distinguishes it with pseudo-transformational leadership.

Whereas social exchange theory posits that the expectation of reciprocity is the key mechanism that motivates people or bonds them together, the value-based perspective highlights the role of value congruence for motivating people and developing cohesive human interactions. This stream of research is supported by the theory of person-organization fit [41]. In other words, employees in an organization are committed to and satisfied with organizations when they believe their organizations are promoting common values [42–44]. Both the theories of PSM and transformational leadership [45] share the underlying ideas that the fit between organization and individual in terms of value pursuit is one of the core antecedents of employee motivation. Additionally, the notion of a normative psychological contract has recently been introduced to capture the value-centered aspect of implicit expectations of employees toward their organization and supervisor [46].

There are many reasons that organizational justice influences employees' work attitudes through the mediating link of intrinsic motivation. Intrinsic motivation refers to "the doing of an activity for its inherent satisfactions rather than for some separable consequence [47] p. 56". While early scholars focused on the interestingness of given task as a facilitator of intrinsic motivation [48], later scholars have explored a wide variety of antecedents of intrinsic motivation, based on self-determination theory [34]. According to self-determination theory, people are best motivated when they perform given tasks driven by the desire for self-realization. People are likely to perceive their actions as based on their own self-determination because of either their innate preferences (they enjoy doing the tasks) or the values that individuals pursue (they believe that doing the tasks is morally right). In this vein, Ren [1] found that employees with the belief that their organizations are pursuing the same values that they appreciate will be more intrinsically motivated than those without such a belief.

The breach of organizational justice will undermine the employee's intrinsic motivation. Given that the issue of justice is related to the fundamental values that most people likely appreciate [21], simply noticing the justice breach experience of coworkers can harm employees' own attitudes. It is very natural to have bad feelings about what is morally wrong and eventually be demotivated intrinsically even if the breach does not entail damages to one's own interest, namely, even if the breach does not affect one's extrinsic motivation. Zapata-Phelan et al. [26] provided a detailed account of why organizational justice affects intrinsic motivation. They begin by positing that positive emotion such as excitement and enthusiasm is a core element of intrinsic emotion, and negative emotions hinder arousal of intrinsic motivation. They go on to argue that a violation of procedural justice breeds negative emotions which eventually hamper intrinsic motivation. Since

treating employees with respect is a norm and widely accepted ethic, the breach of interactional justice will also result in decreased intrinsic motivation. Similarly, Cropanzano, Byrne, Bobocel and Rupp [18] posited that humans have the innate need for self-esteem, belongingness, and morally meaningful existence, and proposed that mistreatment by other people threatens these needs and creates negative psychological reactions. All of these negative psychological reactions to injustice will harm employees' intrinsic motivation.

**Hypothesis 2.** *The effect of organizational justice on job satisfaction is mediated by intrinsic motivation.*

### 2.3. Relative Strengths of Intrinsic and Extrinsic Motivation

Many have found that public employees with stronger PSM showed a higher level of satisfaction, commitment, intention to remain and performance [29,31]. Second, the literature on transformational leadership also attests to the greater importance of intrinsic motivation than extrinsic motivation. Given that transformational leadership is proposed as the style of leadership which leads to performance beyond expectations [39], transformational leadership is assumed to have a greater motivational effect than transactional leadership. Such an assumption has been verified in a variety of empirical studies [16,22,49]. MacKenzie, Podsakoff and Rich [50] found that transformational leadership more significantly affects a salesperson's performance and organizational citizenship behavior than transactional leader behaviors. Similarly, Trottier, Van Wart and Wang [51] found that the effect of transformational leadership on followers' perception of leadership effectiveness was greater than that of transactional leadership in the federal agencies. Transformational leadership motivates employees by appealing to the followers' values, and transactional leadership relies heavily on utilizing one's self-interest. Given this, the greater motivational effect of transformational leadership can be translated into the greater importance of intrinsic motivation on employees' attitudes such as job satisfaction.

**Hypothesis 3.** *The effect of organizational justice on job satisfaction mediated by intrinsic motivation will be greater than the effect mediated by extrinsic motivation.*

Based on the hypotheses, we demonstrated the conceptual model that we test in this study, as shown in Figure 1 below.

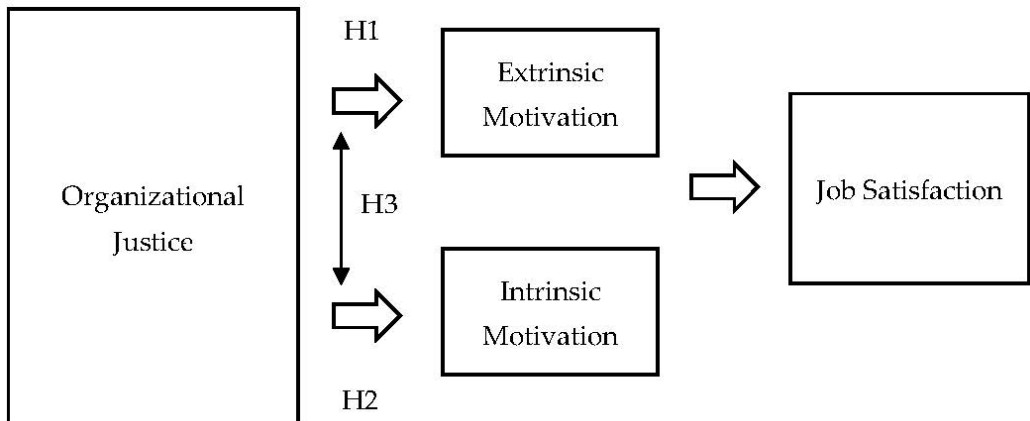

**Figure 1.** Conceptual mode.

## 3. Methods

### 3.1. Data and Sample

In this study, the data from the Office of Personnel Management (OPM) was used for analysis. The OPM collects the survey data from the federal workforce annually, and the responses for the Federal Employee Viewpoint Survey (FEVS) collected in 2019 were used to measure the study variables. FEVS was collected by stratified sampling. Employees are

stratified based on the agency and organizational level. The lowest units had less than 10 individuals. A total of 1,443,152 federal employees were invited to fill out the survey, and 615,395 did, with a response rate of 42.6%. Among the respondents, only those who answered all the questions used to measure study variables were included in the final sample, which made the final sample size of 409,332.

Among the respondents, 70.6% were either non-supervisors or team leaders, and 21.5% were supervisors/managers/executives. Male employees constitute 51.3% of the total respondents, with females at 37.7%, and 11% were non-responses. For ethnicity, 58.3% were white. For organizational tenure, 36.6% were seniority of ten years or fewer, 31.1% were between 10 and 20 years and 23.7% were more than 20 years.

### 3.2. Analysis

The purpose of this study is to examine the mediating effects of intrinsic and extrinsic motivation to the relationship between organizational justice and work attitudes and compare the relative size of the effects between intrinsic and extrinsic paths. To examine the mediating effects, this study used structural equation analysis.

For the analysis, first of all, confirmatory factor analysis (CFA) will be conducted to examine the validity of the measurement model. This will help investigate whether common source bias is severe enough to distort the result of the analysis. Along with CFA, Harman's one-factor test will also be conducted. Second, structural equation analysis will be conducted to examine the proposed hypotheses. In the past, mediation effect was examined using the methods suggested by Baron and Kenny [52]. However, as indicated in a more recent study [25], structural equation modeling is a more advanced and better method for investigating mediation effect.

### 3.3. Measurement

This study uses the data from the Federal Employee Viewpoint Survey (FEVS), the annual survey conducted by the Office of Personnel Management. This study contains many important aspects of the attitudes and organizational life of federal employees. According to Fernandez, Resh, Moldogaziev and Oberfield [53], many public management scholars have used this data to examine various topics of public management including organizational justice. Since the FEVS is secondary, Fernandez et al. [53] advised that researchers have to provide a robust theoretical basis for constructing the measurements. In this study, the selection of measurements is based on referring to the prior studies which used the same constructs, or a thorough theoretical discussion of the constructs.

As the target variable, this study used the overall satisfaction of employees. This construct is used because satisfaction is one of the most widely used measures of employee attitudes. In many studies, along with commitment, satisfaction has been used as a praxis of motivation in the management literature [54]. Scholars have proved that employees' satisfaction is an important work outcome which is also closely related to job performance [55] and commitment [56]. This study used 'overall' satisfaction, without specifying the foci and factors of satisfaction, because this study intended to capture a general indication of how well employees fit in the organizational environment and perform as a contributor of accomplishing goals. To measure this variable, the following three questions were used: "Considering everything, how satisfied are you with your job?"; "Considering everything, how satisfied are you with your organization?"; "I recommend my organization as a good place to work". These questions were the same items used in the work of Trottier et al. [57] for capturing follower's satisfaction. The Cronbach's alpha reliability score was 913.

To measure independent variables, this study included the three components of organizational justice in the proposed model. The three components are distributive justice, procedural justice and interactional justice [38]. Distributive justice is the component first introduced in the literature. This concept of justice refers to the perceived fairness of the result of distributing organizational rewards [58]. Adams [3] suggested that the perception of fair distribution depends on the ratio of one's contributions to one's reward outcomes,

and comparison of this ratio of one person with that of others. Applying this framework into organizational context, one's contribution can be translated to 'work performance'.

Thus, in sum, distributive justice can be measured by questions asking whether one believes that organizational rewards are distributive by performance or merit, based on the measurement scale used in Kim and Park [58]. The following three questions were used to measure distributive justice: "My performance appraisal is a fair reflection of my performance (this is a fundamental condition that must be met in order for a performance-based distribution to be truly fair)"; "Promotions in my work unit are based on merit"; "Awards in my work unit depend on how well employees perform their jobs"; "Pay raises depend on how well employees perform their jobs". The reliability for this variable was 846.

Procedural justice was first introduced into the concept of organizational justice by Thibaut and Walker [59]. Later, Leventhal [60] elaborated the construct. According to his formulation, some of the conditions for procedural justice are that procedures should be applied consistently across people and across time, be free from bias, have some mechanism to correct flawed practices and conform to prevailing standards of ethics. The following three questions seem to capture some of these conditions: "I can disclose suspected violation of any law, rule, or regulation without fear of appraisal (any flawed practices can be corrected by internal report)"; "Arbitrary action, personal favoritism, and coercion for partisan political purposes are not tolerated (the rules are always applied without exceptions)"; "Prohibited Personnel Practices (for example, illegally discriminating for or against any employee/applicant, obstructing a person's right to compete for employment, knowingly violating veterans' preference requirements) are not tolerated". The reliability alpha for this variable was 863.

Interactional justice focuses on the perceived fairness of interpersonal treatment people received in organizational contexts. This concept focuses on the extent to which people are treated with dignity and respect and not discriminated based on inappropriate criteria [31]. The following four questions were used to measure this variable, "My supervisor is committed to a workforce representative of all segments of society (not discriminated)"; "My supervisor listens to what I have to say (respect)"; "My supervisor treats me with respect (respect)"; "Supervisors work well with employees of different backgrounds (not discriminated)". The reliability was 894.

Two mediating variables were intrinsic motivation and extrinsic motivation. We used the scale found in Lee [41]. First, to measure intrinsic motivation, the following three questions were used: "My work gives me a feeling of personal accomplishment"; "I like the kind of work I do"; "I know how my work relates to the agency's goals and priorities"; "My talents are well used in the workplace". The first three questions were used to measure intrinsic motivation in a prior study [14]. The last question was added because it seems closely associated with an intrinsic motivation based on the formulation of cognitive evaluation theory (CET) [60]. According to CET, those who have a strong sense of positive self, for instance, positive self-image related to one's competence, will have strong intrinsic motivation. In this vein, Cropanzano et al. [18] also argued that self-esteem, competence, and sense of belonging are essential elements of intrinsic motivation. Those who realize that their talents are being well used in the workplace are likely to have high self-esteem as a result and grow a strong sense of competence, and a sense of being accepted by the organization, and hence are more likely to show strong intrinsic motivation. The reliability was 830.

To measure another mediating variable, extrinsic motivation, this study focuses on the formal definition, "whenever an activity is done in order to attain some separable outcome" [60] (p. 60). Both tangible reward (e.g., cash bonus or pay raise) and intangible reward (e.g., recognition) constitute extrinsic factors. Based on this definition, to measure the extent to which one is extrinsically motivated, this study selected the questions asking whether one received extrinsic rewards for doing good work. These questions were, "Employees are recognized for providing high-quality products and services"; "Creativity

and innovation are rewarded"; "In my work unit, differences in performance are recognized in a meaningful way"; "How satisfied are you with the recognition you receive for doing a good job?". The reliability alpha was 910.

## 4. Results

### 4.1. Measurement Model

All the variables in this study were measured using the data from a single source, the 2019 FEVS. Since this data is obtained from a self-reported survey, the respondents' psychological desire for maintaining a consistent tone while answering multiple questions may have inflated the associations among the study variables, which is called "common source bias". Since it is not desirable to statistically eliminate this bias [27], it is a necessary process to examine whether the problem is severe enough to cause bias in the results.

Podsakoff, MacKenzie, Lee, and Podsakoff [61] proposed several methods for detecting the severity of common method bias. This study will use Harman's one-factor test and CFA. First, Harman's one-factor test suggests that we run an exploratory factor analysis, and if a single factor emerges and that factor explains most of the total variance, then it indicates that the problem of common source bias is significant enough to invalidate the results. In this study, three factors are identified instead of a single factor, and the factor with the greatest loading explains only 32.01% of the total variance. Secondly, this study also conducted CFA for two different models: one is the proposed six-factor model, and the other is the one-factor model that represents the condition in which respondents' consistent tone actually drives their answers to all the questions. The result of CFA indicates that the six-factor model explains the data much better than the one-factor model. The fit indices for the proposed six-factor model were all within the acceptable range except for Chi-square (Chi-square = 580,992.175, $p < 0.000$, SRMR = 0.071, NFI = 0.923, RFI = 0.908, IFI = 0.923, TLI = 0.908, CFI = 0.923, RMSEA = 0.086), while the fit indices for the one-factor model were all below the standards (Chi-square = 1,627,763.016, $p < 0.000$, SRMR = 0.069, NFI = 0.785, RFI = 0.762, IFI = 0.785, TLI = 0.762, CFI = 0.785, RMSEA = 0.138). Based on the results of these two tests, it is concluded that the common source bias is not severe enough to invalidate the results. Table 1 presents the descriptive statistics and correlation matrix.

**Table 1.** Descriptive statistics and correlation matrix.

|  | Mean | Std. Dev. | Intrinsic Motivation | Extrinsic Motivation | Overall Satisfaction | Procedural Justice | Distributive Justice |
|---|---|---|---|---|---|---|---|
| Intrinsic Motivation | 4.052 | 0.794 |  |  |  |  |  |
| Extrinsic Motivation | 3.381 | 1.042 | 0.645 ** |  |  |  |  |
| Overall Satisfaction | 3.785 | 0.989 | 0.744 ** | 0.766 ** |  |  |  |
| Procedural Justice | 3.748 | 1.031 | 0.593 ** | 0.740 ** | 0.706 ** |  |  |
| Distributive Justice | 3.316 | 0.990 | 0.612 ** | 0.867 ** | 0.708 ** | 0.723 ** |  |
| Interactional Justice | 4.076 | 0.875 | 0.590 ** | 0.700 ** | 0.675 ** | 0.706 ** | 0.670 ** |

** $p < 0.01$.

### 4.2. Structural Model

To discuss the fit of the proposed model, overall, the fit indices indicate an acceptable fit (Chi-square = 598,636.497, $p < 0.000$, SRMR = 0.071, NFI = 0.921, RFI = 0.908, IFI = 0.921, TLI = 0.908, CFI = 0.921, RMSEA = 0.086). Although Chi-square was statistically significant, given the extremely large sample size, the significant Chi-square alone provides no robust basis for entirely dismissing the fit of the proposed model. Although some of

the indices indicate a barely acceptable fit (e.g., RFI, TLI), overall, they indicate a generally acceptable fit.

To examine the individual path coefficients, all coefficients indicate a positively significant association among the variables. Among these, the association between distributive justice and extrinsic motivation showed the greatest effect size (0.886), and the association between interactional justice and extrinsic motivation was the weakest (0.041). Figure 2 indicates the path coefficients. Hence, Hypothesis 1 and 2 are supported.

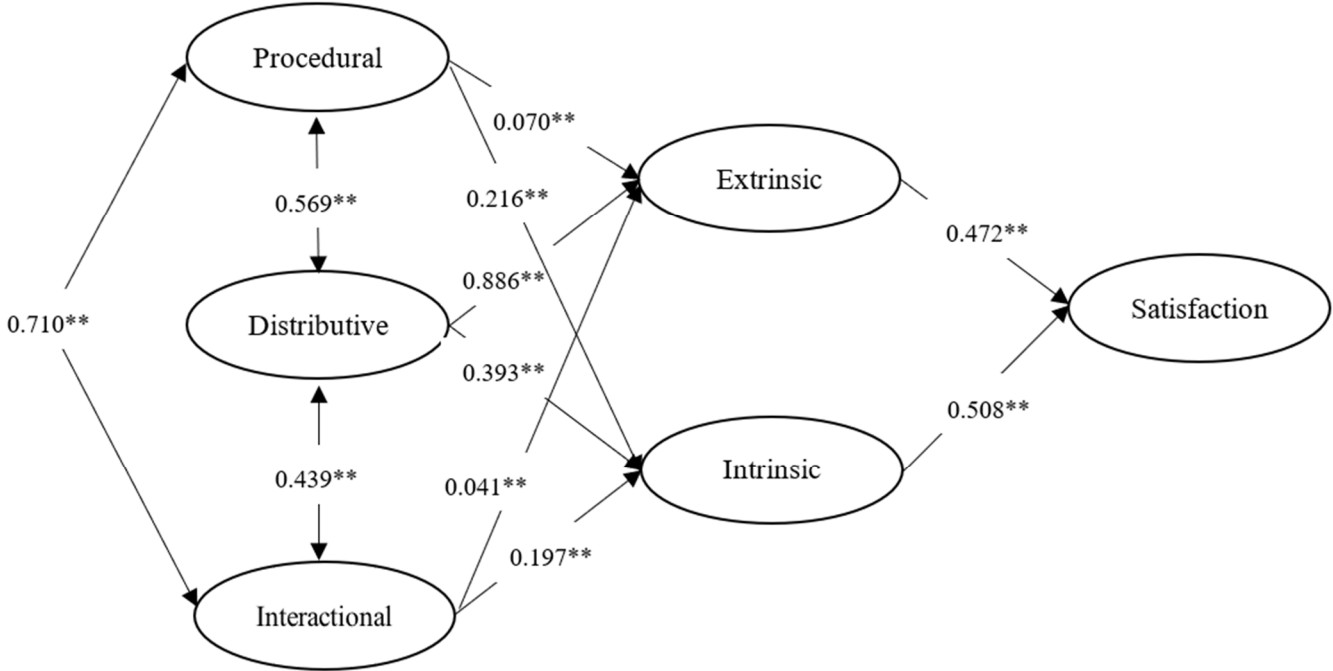

**Figure 2.** Path coefficients. (**denotes the significance at 0.1).

The result of testing indirect effects is the following. The indirect effect of distributive justice through intrinsic motivation was 0.200, while the indirect effect through extrinsic motivation was 0.418. The indirect effect of procedural justice through intrinsic motivation was 0.110, and through extrinsic motivation, it was 0.033. The indirect effect of interactional justice through intrinsic motivation was 0.100, while the indirect effect through extrinsic motivation was 0.019. To sum all of these indirect effects, the total indirect effect through intrinsic motivation was 0.410, while the total indirect effect through extrinsic motivation was 0.470. Table 2 presents the indirect effects. The result shows that the total effects are comparable to one another, but the difference of indirect effects among individual elements of organizational justice are considerable (Hypothesis 3).

**Table 2.** Indirect effects on job satisfaction.

| | Mediating Variable | |
|---|---|---|
| **Independent Variable** | **Intrinsic Motivation** | **Extrinsic Motivation** |
| Distributive Justice | 0.200 | 0.418 |
| Procedural Justice | 0.110 | 0.033 |
| Interactional Justice | 0.100 | 0.019 |
| Total Effect | 0.410 | 0.470 |

## 5. Discussions and Conclusions

### 5.1. Theoretical Implications

Our study aimed to investigate the mediating role of intrinsic motivation in the relationship between organizational justice and work attitudes. Consistent with our initial

hypothesis, our results revealed a significant indirect effect of intrinsic motivation on employee satisfaction, highlighting the importance of this mechanism as a pathway through which organizational justice enhances work attitudes. Importantly, our findings also align with prior research showing that employees not only expect to fulfill a calculative psychological contract but also a normative one [46]. Together, these results underscore the complex nature of employee motivation and the need for organizations to consider multiple factors in order to promote positive work attitudes.

One of the key contributions of this study is that it offers a comprehensive view of the motivating mechanisms of organizational justice by examining the mediating roles of both intrinsic and extrinsic motivation. Previous studies have typically focused on one type of motivation, but our analysis shows how both types of motivation work together to drive employee behavior and attitudes. By accounting for these complex relationships, our study offers new insights into how organizations can foster a more motivated and engaged workforce. Particularly, we helped understand not only the relative importance among the elements of organizational justice, but also the significance of the two causal paths that link organizational justice to intrinsic motivation and extrinsic motivation, respectively, which advances our understanding of the motivational mechanism that has not been covered in prior research by Deschamps et al. [22].

In the following, we provide a detailed discussion of several noteworthy findings that emerged from our analysis. First of all, the sum of the indirect effects via intrinsic motivation across three components of organizational justice (0.410) was comparable to the sum of the indirect effects via extrinsic motivation (0.470). This indicates that the overall influence of organizational justice on work attitudes through intrinsic motivation is just as much as its influences on the attitudes via extrinsic motivation. This result illuminates the hidden impact of organizational justice that has hardly been researched by previous scholars. Given that intrinsic motivation is a more powerful driver for work attitudes and work efforts than extrinsic motivation [60,61], which is also indicated in the result of this study, the path coefficient for intrinsic motivation-overall satisfaction (0.508) was a bit higher than that for extrinsic motivation-overall satisfaction (0.472), meaning the impact of the breach of organizational justice extends far beyond what previous research have predicted based on social exchange theory.

Aryee et al. [24] already found that organization justice is significantly related to intrinsic motivation. However, the findings of this study are not redundant. They do add unique contributions to the literature beyond their study by presenting the results of comparing the relative strengths of each element of organizational justice for predicting intrinsic and extrinsic motivation, which has largely been ignored in prior studies. To discuss each result in detail, first, the effect of procedural justice on overall satisfaction was explained largely by the indirect effect through intrinsic motivation, rather than through extrinsic motivation. The effect size was greater for the path via intrinsic motivation, compared to the effect size of the path via extrinsic motivation. This result indicates that the issue of procedural justice is closely related to the value congruence of individual employees, rather than as simply a matter of a loss in employees' tangible benefits. It implies that the breach of procedural justice not only hamper employees' extrinsic motivation, but also their intrinsic motivation because the breach makes employees doubt the congruence in terms of the value pursuits between their organizations and their own.

Second, the same was true for interactional justice. The indirect effect of interactional justice was also greater for the path via intrinsic motivation than that of extrinsic motivation. This also implies that the major reason that organizational justice is associated with workers' attitudes are that the breach of interactional justice makes employees develop negative beliefs toward their organization's commitment to the noble values that they highly appreciate. Moreover, the breach of interactional justice of treating them with disrespect may lower their own evaluation of self-image, thereby reducing intrinsic motivation.

Third, to the contrary, the relative strength (effect size) of the indirect effects was the opposite as that of distributive justice. In other words, the magnitude of the indirect effect

of distributive justice was greater for the path via extrinsic motivation than the path via intrinsic motivation. In other words, the breach of distributive justice was manifested mainly through decreased extrinsic motivation, rather than decreased intrinsic motivation. Of course, the effect size of distributive justice through the mediating linkage of intrinsic motivation was also considerable and statistically significant. However, a greater amount of variance was explained by extrinsic motivation linkage. This result makes sense because distributive justice is by definition closely related to the number of extrinsic rewards employees receive as the outcome of the distribution process, while other two components of justice, namely, procedural and interactional justice are not directly linked to the extrinsic rewards.

*5.2. Practical Implications*

These findings provide a number of practical implications. The study showed that promoting organizational justice is one of the important ways to increase workers' motivation and elicit positive work attitudes. Previous scholars have identified some antecedents to intrinsic motivation. The examples of these determinants are task characteristics [48] and managers' transformational leadership [62]. The result of this study adds organizational justice to the list of antecedents of intrinsic motivation. In order to promote intrinsic motivation of employees, the management may have to pay attention to the transparency of decision makings, thereby improving employees' perception of organizational justice. This can be done by explaining why certain procedures are enacted and providing information as to why managers have to make a particular decision as to the distribution of organizational rewards. In addition, organizations may have to consider utilizing formal training programs to increase the awareness and sensitivity of managers as to the justice issues within their organization [63].

The second important implication is that none of the elements of organizational justice should be overlooked. Distributive justice is the element first identified among the three elements of organizational justice, and the other two were added later [16]. However, this does not discount the importance of the other two elements at all. In fact, some might argue that the greater importance of the other two elements, procedural and interactional justice, for intrinsic motivation implies that they should be treated with more emphasis than distributive justice, since the effect of intrinsic motivation is generally stronger [27] and longer lasting [64] than extrinsic motivation.

Third, this finding also has some implications for performance appraisal. Many have reported that there exists the tendency of managers to give overly generous performance grades to all of their subordinates. This leniency effect may seem to be a solution leading to win-win situations for all employees. However, the result of this study shows that it may eventually harm the motivation of its employees. The leniency effect may increase extrinsic motivation by positively affecting the perception of distributive justice. However, there should be numerous employees who believe that this practice is not just and of integrity. These employees are likely to experience a decline in their perceptions of procedural justice, which eventually undermines their intrinsic motivation. On a similar note, this finding also indicates a warning against favoritism. Giving favor to some employees might seem to increase the extrinsic motivation of those beneficiaries by increasing their perception of interactional justice. However, it is not only general other people but also those beneficiaries who will experience the decrease in intrinsic motivation in the long term, because favoritism will give rise to the uncomfortable feeling coming from the breach of procedural and distributive justice. This is very likely to be the case given that even the beneficiary would think that there is no guarantee that they will continue to receive the favorable treatment.

Lastly, the finding indicates that both intrinsic and extrinsic motivation positively influenced job satisfaction, and the effect of intrinsic motivation was slightly greater. This implies that managers need to pay attention to both the intrinsic and extrinsic motivations of their subordinates. Whereas social exchange theorists prescribe the use of perceived or-

ganizational support and highlight the importance of leader-member exchange (LMX) [65] as a motivating mechanism, the value-based perspective emphasizes the hiring of those with compatible values [41] and transformational leadership [45]. In fact, this prescription provides the basis for managerial advice to take a balanced approach to motivate employees, which is very similar to those suggested in the leadership literature. For instance, O'Shea, Foti, Hauenstein and Bycio [57] suggest that optimal leadership is neither extreme transformational nor extreme transactional leadership; rather, it is the leadership that embraces some element of transformational leadership and some aspects of transactional leadership. Likewise, managers need to attempt to find an optimal mix of focusing both on intrinsic and extrinsic motivation.

*5.3. Implications for Sustainability*

Organizational justice has been treated important for promoting sustainability in organizations [66,67]. This study adds value to the literature by providing some new insights into the motivational effects of organizational justice. As noted in Table 2, the individual elements of organizational justice have differential effects on intrinsic motivation. While distributive justice more strongly affected extrinsic motivation than intrinsic motivation, procedural and interactional justice had stronger effects on intrinsic motivation than extrinsic motivation. This result shows that in order to achieve organizational sustainability via intrinsic motivation, a manager should pay attention not only to increasing distributive justice, but also to promote procedural justice and interactional justice.

In some workplace situations, managers encounter the dilemma between distributive and procedural justice. For example, managers often determine performance evaluation of their subordinates based on rotation such that those employees nearing promotion receive higher ratings. Our results show that other employees might view this as a sign of unjust organizational practice given that rotation is not a proper procedure for performance evaluation. In this case, the practice of rotation can hamper intrinsic motivation and in turn undermine job satisfaction.

Of course, this can be an established group norm in some organizations. If so, the result of this study may be inapplicable, such that rather, in practice, sudden change to this established practice can be viewed as an unjust managerial decision, which decreases intrinsic motivation and sustainability.

Therefore, managers need to be cautious in implementing the results suggested by this study. However, at the very least, the results of this study provide managers with a caveat that if a majority of employees believe that the practice of rotation is unjust, they need to cease the rotation of performance ratings to maintain a proper level of intrinsic motivation. This change may be beneficial for organizational sustainability in the long run.

## 6. Limitation and Avenue for Future Research

The contribution of this study notwithstanding, this study has some limitations as well. Future researchers may have to consider these shortcomings in addressing similar research questions. First, this study uses data from the self-reported survey. Hence, although the severity of the common source bias has been tested, the possibility of that bias is not completely eliminated. Thus, later researchers may have to complement their work with the use of secondary data to test the effect of organizational justice on task, job and organizational performance, which are other types of indicators of organizational outcomes. Second, this study used cross-sectional data. Hence, it is hard to convincingly infer causal relationships among the study variables. Future researchers may have to use longitudinal data to examine causal relationships. Third, as a target variable, this study used only overall employees' satisfaction. Although satisfaction is a typical and probably the most important indicator of positive employee attitude, it may be possible that the relative strengths between intrinsic motivation and extrinsic motivation in explaining the target variables may vary by types of work attitudes. Thus, future researchers may have to test the effect of organizational justice against more diverse work attitudinal variables, such as

work effort and task performance. Lastly, the relative importance of intrinsic and extrinsic motivation may differ depending on individual differences, which are omitted variables in this study. Thus, one may have to confirm or challenge the results of this study by conducting a similar study in the private sector contexts, in which an employee is inclined to extrinsic motivation, rather than intrinsic motivation.

**Author Contributions:** H.-W.L. planned this study and drafted a manuscript. D.-Y.R. helped analyze the data and revised the manuscript. All authors have read and agreed to the published version of the manuscript.

**Funding:** This research was supported by the Hallym University Research Fund, HRF-202212-001.

**Institutional Review Board Statement:** The IRB approval was obtained by the U.S. federal government website.

**Informed Consent Statement:** Informed consent was obtained by the U.S. federal government.

**Data Availability Statement:** The data for this study is available online.

**Conflicts of Interest:** The authors declare no conflict of interest.

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
