# Peer review of "Effects of Organizational Justice on Employee Satisfaction: Integrating the Exchange and the Value-Based Perspectives"

_sustainability, doi:10.3390/su15075993_

Round 1

Reviewer 1 Report

The topic selection of the thesis has strong practical and theoretical significance, and it has even more special significance in today's emphasis on pluralistic governance. The thesis hypothesis has a more appropriate theoretical basis, the research method is effective, the research model is reliable, the statistical analysis process is clear, the result is clear, and the discussion of the conclusion has a relatively clear and clear point of view. It is suggested to add more recent literature and review for the relative study are.

Suggested revisions before consideration for publication.

Author Response

We strengthened the theory and literature review section according to your suggestion.

Reviewer 2 Report

I found this paper has attempted to provide empirical findings for public servants by adopting or replicating a study by Fernandez et al (2015). However, the methodology and the selection of your respondents have to be further clarified. Please justify your ethical conduct during data collection process and selection of your respondents.

The LR has to be updated as the latest is only from 3 studies in 2015 while others are still relevant but very old as compared to being published in the year 2023. Therefore what happened in 2016-2022 and the implications for the theoretical contributions of the study? Please justify the significance of your study in your public administration policy in your country's context in order to strengthen your problem statement and contributions to your study. Not so sure how this topic is relevant or related to sustainability where your issue and discussion could be strengthened for this area.

Author Response

I found this paper has attempted to provide empirical findings for public servants by adopting or replicating a study by Fernandez et al (2015).

: With all due respect, we would like to make it clear that the study under review is not a reproduction or adoption of Fernandez et al. (2015). This study only used FEVS data, and Fernandez and his colleagues critically reviewed studies using this data.

However, the methodology and the selection of your respondents have to be further clarified. Please justify your ethical conduct during data collection process and selection of your respondents.

: This study utilized the secondary data collected by the third party (i.e., The Office of Personnel Management in the U.S. Federal Government).

A brief explanation of the data collection process is the following which is briefly explained in the text as well.

  1. Stratify individuals based on the lowest desired work unit or “level” identified by the agency.
  2. Identify strata with less than 10 individuals and roll these up into the next-highest applicable stratum. This rolling up was performed because even if a 100% response rate were achieved, a work unit of 10 would be too small to receive a report. If there is no applicable higher level within the agency structure, the stratum is left as is.
  3. As individuals in senior leader positions (e.g., SES or equivalent) constitute a rare subgroup of analytic interest, place them into a separate stratum to ensure they are sufficiently represented in the agency sample.
  4. Once the final stratification boundaries were set, the sampling proportion was assigned based on the size of the stratum and the goal of attaining at least 10 respondents. We assumed a conservative 30% response rate. Exceptions to this rule were any strata in small agencies and the SES strata. These were censused. As seen in Table 1, the minimum sampling proportion was 25%; thus, each employee had at least a one in four chance of being selected to participate.
  5. After the necessary sample size is determined, the agency’s ratio of employees to be sampled was examined. If 75% or more of the workforce was to be sampled, a census of the agency was conducted instead.

As for the ethical conduct, we do not have specific details. However, we assume that the data collection followed the ethical standards as it was collected by the U.S. federal government.

More information as to the FEVS 2015 can be found at the following website (https://www.opm.gov/fevs/reports/technical-reports/technical-report/technical-report/2015/2015-technical-report.pdf).

The LR has to be updated as the latest is only from 3 studies in 2015 while others are still relevant but very old as compared to being published in the year 2023. Therefore what happened in 2016-2022 and the implications for the theoretical contributions of the study?

: We strengthened the theory and LR section by citing more recent studies and the discussion section as well (p.10). But we would also like you to understand that this study used the established theoretical perspectives in the literature, which makes it more appropriate to cite the original(old) studies.

Please justify the significance of your study in your public administration policy in your country's context in order to strengthen your problem statement and contributions to your study.

: The authors are not from the country from which the data used in this study is collected. We instead mentioned on the universal relevance of the value-based perspective in managing public employee’s motivation (p.3).

Not so sure how this topic is relevant or related to sustainability where your issue and discussion could be strengthened for this area.

: Organizational justice has been a popular topic of interest among sustainable management scholars whose research has been published this journal. The implication of this study to sustainability is explained in 5.3. at the end of the article (p. 12-13). To make it clearer, we added a few sentences in the abstract and others (e.g., p.12). I would like you to consider that this article is submitted to the special issue regarding sustainable organization, rather than sustainability in general.

Reviewer 3 Report

The article has a good topic and the authors have produced a clear text.

1. this study examined the mediating effects of both intrinsic and extrinsic  motivation linking between organizational justice and employee satisfaction

  • In my opinion, both the hypothesis testing and the Results section focus on this proposal.

2. I think the topic is not so original but relevant. Actually, there is an extensive research about Public Service Motivation.

3. I think this manuscript addresses a specific gap in the field. The author state:

….more investigation is still needed because Aryee et al. (2015) tested the effect of overall justice, rather than the impacts of the three components of organizational justice separately.

“Notably, the extensive research on public service motivation (PSM) has proved that intrinsic factors are far more critical for public employees’ motivation than extrinsic rewards (Crewson, 1997). Hence, this study will add an important theoretical basis for the imperative of promoting organizational justice.”

4. Public service motivation is a topic is an issue that has been growing significantly in recent years.

Source: Scopus.

And the author state:

“..add unique contributions to the literature beyond their study by presenting the results of com paring the relative strengths of each element of organizational justice for predicting intrinsic and extrinsic motivation, which has largely been ignored in the prior studies”

However, I believe it is important to review the current scientific production on the subject in order to validate its relevance in recent years.

5. I think the sample is appropriated as well as the CFA technic data analysis. The only concern may be regarding the measurement scales although the authors state:

“According to Fernandez, Resh, Moldogaziev, & Oberfield (2015), many public management  scholars have used this data to examine various topics of public management including organizational justice. Since the FEVS is the secondary, Fernandez et al. (2015) advised that researchers have to provide a robust theoretical basis for constructing the measurements. In this study, selection of measurements is based on referring to the prior studies which used the same constructs, or a thorough theoretical discussion of the constructs “

  • I would then request that the authors include the bibliographic references of previous studies.
  • Also, they must include the information regarding the measurement scales: reliability coefficients.
  • It is necessary to a description of the scale of measurement of satisfaction

6. According to the aim of the research: “To fill that gap, this study examined the mediating effects of both intrinsic and extrinsic  motivation linking between organizational justice and employee satisfaction”

There is an inaccuracy in this paragraph: “The purpose of this study was to illuminate the indirect effect of intrinsic motivation mediating between organizational justice and work attitudes”. So the authors should improve this paragraph.

  • I recommend that the authors review the presentation of results according to the aim of the research and the hypotheses.

7. References are appropriate. But I consider they should to use more actual references. There are more than 150 research papers to be published between 2021 and 2023.

8. Additional comments on the tables and figures:

  • Tables and figures: Must include the source.
  • Figure: May be it is necessary to include the word “Satisfaction”. And review the Path Coefficient

Other:

In Theory and hypothesis section:

The information in lines 210 to 220 is not relevant.

Author Response

The article has a good topic and the authors have produced a clear text.

  1. this study examined the mediating effects of both intrinsic and extrinsic  motivation linking between organizational justice and employee satisfaction
  • In my opinion, both the hypothesis testing and the Results section focus on this proposal.

: To be honest, we do not understand your request as to the above comment. If you can elaborate your request, we will address accordingly in the next round of review process.

  1. I think the topic is not so original but relevant. Actually, there is an extensive research about Public Service Motivation.

: We strengthened the theory and literature review sections by citing more research.

  1. I think this manuscript addresses a specific gap in the field. The author state:

….more investigation is still needed because Aryee et al. (2015) tested the effect of overall justice, rather than the impacts of the three components of organizational justice separately.

“Notably, the extensive research on public service motivation (PSM) has proved that intrinsic factors are far more critical for public employees’ motivation than extrinsic rewards (Crewson, 1997). Hence, this study will add an important theoretical basis for the imperative of promoting organizational justice.”

: We would like to appreciate your complement.

  1. Public service motivation is a topic is an issue that has been growing significantly in recent years.

Source: Scopus.

And the author state:

“..add unique contributions to the literature beyond their study by presenting the results of comparing the relative strengths of each element of organizational justice for predicting intrinsic and extrinsic motivation, which has largely been ignored in the prior studies”

However, I believe it is important to review the current scientific production on the subject in order to validate its relevance in recent years.

: We strengthened the theory and literature review sections by citing more recent studies. Also, we discussed the relevance and contribution of this study to the literature in light of new studies adopted.

  1. I think the sample is appropriated as well as the CFA technic data analysis. The only concern may be regarding the measurement scales although the authors state:

“According to Fernandez, Resh, Moldogaziev, & Oberfield (2015), many public management scholars have used this data to examine various topics of public management including organizational justice. Since the FEVS is the secondary, Fernandez et al. (2015) advised that researchers have to provide a robust theoretical basis for constructing the measurements. In this study, selection of measurements is based on referring to the prior studies which used the same constructs, or a thorough theoretical discussion of the constructs “

  • I would then request that the authors include the bibliographic references of previous studies.

: We assume that you are referring to the previous studies which used the measures in this study. The bibliographic references are added where applicable.

  • Also, they must include the information regarding the measurement scales: reliability coefficients.

: Cronbach alpha is the reliability coefficient which are already at the end of the descriptions of each measure.

  • It is necessary to a description of the scale of measurement of satisfaction

: That is on the very first line of page 7.

  1. According to the aim of the research: “To fill that gap, this study examined the mediating effects of both intrinsic and extrinsic motivation linking between organizational justice and employee satisfaction”

There is an inaccuracy in this paragraph: “The purpose of this study was to illuminate the indirect effect of intrinsic motivation mediating between organizational justice and work attitudes”. So the authors should improve this paragraph.

  • I recommend that the authors review the presentation of results according to the aim of the research and the hypotheses.

: In response to your request, we added extra sentences explaining what the results mean.

  1. References are appropriate. But I consider they should to use more actual references. There are more than 150 research papers to be published between 2021 and 2023.

: We strengthened the theory and literature review sections by citing more recent studies.

  1. Additional comments on the tables and figures:
  • Tables and figures: Must include the source.

: The tables and figures are created by our own based on the results of analysis. We examined other equivalent studies (SEM analysis) published in this journal to find that they do not indicate the sources. If you are sure that the sources need to be indicated, we will do that in the next round.

  • Figure: Maybe it is necessary to include the word “Satisfaction”. And review the Path Coefficient

: The texts in the circles of Figure 1 were covered while editing. We have revised it.

Other:

In Theory and hypothesis section:

The information in lines 210 to 220 is not relevant.

: The part dealing with the distinction between PSM and intrinsic motivation has been eliminated.

Reviewer 4 Report

Dear authors,

I appreciate having the opportunity to review the manuscript entitled “Effects of Organizational Justice on Employee Satisfaction: Integrating Social Exchange and Value-based Perspective” (Manuscript ID: sustainability-2161593). I enjoyed your paper.

The current study investigated the mediating influences of both intrinsic and extrinsic motivation linking between organizational justice and employee satisfaction. The results of the analysis shows that the indirect mediating effects of intrinsic motivation were greater than those of extrinsic motivation as for procedural and interactional justice, while the mediating influence via extrinsic motivation was greater when it comes to distributive justice. Also, the sum of the indirect effects of intrinsic motivation was comparable to that of extrinsic motivation.

I believe that the current manuscript is enough to be published at Sustainability, based on its adequate capturing research questions, transforming those into hypotheses, testing the hypotheses with good methodology, and presenting those clearly. However, I want to provide my opinions to contribute to improving this paper.

- I think that the overall structure and writing of introduction part are clear and well-aligned, thus it is easy for me to catch what the research questions and strategies to deal with of this paper.

- This paper not only dealt with interesting phenomena, but also provided adequate theoretical background and support for the development of its hypotheses. This is the strong point of this paper.

- However, unfortunately, I think that the current research needs to add some more recent papers on organizational justice, satisfaction, and motivation. I suggest the authors to revise the manuscript by adding the papers.

I wish these comments may help you to improve your paper. Good luck.

Author Response

We greatly appreciate you complement on the strengths of this paper. Responding to your request, we strengthened the theory and literature review sections citing more recent studies on the topic.

Round 2

Reviewer 2 Report

I appreciate your effort to respond to my previous comments with your additional input and amendments in the draft as to improve the quality and clarity. I found it is more appealing to justify your usage of secondary data from OPM and FEVS 2015 for your methodology and also with the updated references to support your explanations and justification in several parts of your text.

Although it is suffice to answer my previous comments, I have several recommendations and additional feedback after revising your updated draft. Hopefully:

1) You may improve your writing structure/coherent in intro to justify your combination of SET and value-based perspective. You may make a little adjustment in paragraph three and four regarding intrinsic and extrinsic motivation as your value-based perspective by reflecting to your title. I found you have mentioned about values in both para which is good, however I recommend that you may mention the consistent term such as "value-based perspective" which appeared in your title at this part? So that the reader can notice your contribution/justification in combining two theories consistently in your title, abstract, keywords and intro. I found your LR has appeared clearly the subsections for both which is good. It is just the improvement in writing style to make your intro, abstract and keywords will be relevant and easy to read. It is nicer/usable if you may have your figure for conceptual/theoretical framework earlier at LR section, although later I can see it with results in your Figure 1 in findings section. 

2)     Your methodology is improved, however I recommend that you may avoid starting your sentence with numbers especially in reporting your data and sample (3.1). You may start with "A total of..." instead. Check spelling Measurement (3.3).

3) For analysis (3.2), you may check the latest references to report/write your SEM analyses and findings, and not only referring to Baron & Kenny method and Zhao et al (2010).  I recommend you may refer to Table 1 of H. Kang, J.-W. Ahn (2021) from Asian Nursing Research 15 pp157-162, or any other SEM articles. This is important to make sure that all your results, tables and figures in Section 4 are suffice and relevant to report measurement and structural models. Perhaps you may consider to add Sobel test or bootstrapping to justify your mediation analysis, since you mentioned one of your contributions is testing causal effect right? I am more familiar with PLS-SEM where even the measurement model will be reported differently than your approach. Therefore, it is worth to check your report writing with the updated articles in reporting SEM.

4) Finally, it is also worth if you can justify why you referred and used data of FEVS in the year of 2015, rather than the latest ones? I checked the link and found 2016-2019 and not sure about, 2020-2021 may not available because of pandemic etc? I suspect that is why initially I thought that your study was similar to Fernandez 2015, since you may get access to the latest reports for the recent years, nevertheless you mentioned it was collected in 2015 (which was dated for primary data). Now I possibly understand more about your secondary data, perhaps it is  just that you have to clarify why you used 2015.  All the best. 

Author Response

We appreciate your valuable comments helpful for us to improve the manuscript. However, we regret that we were unable to accommodate all your comments due to the short time given to us for revision.

  • You may improve your writing structure/coherent in intro to justify your combination of SET and value-based perspective. You may make a little adjustment in paragraph three and four regarding intrinsic and extrinsic motivation as your value-based perspective by reflecting to your title. I found you have mentioned about values in both para which is good, however I recommend that you may mention the consistent term such as "value-based perspective" which appeared in your title at this part? So that the reader can notice your contribution/justification in combining two theories consistently in your title, abstract, keywords and intro.

: In response to your request, we added the word, ‘value-based perspective,’ to make it clearer (abstract and p.2).

  • I found your LR has appeared clearly the subsections for both which is good. It is just the improvement in writing style to make your intro, abstract and keywords will be relevant and easy to read. It is nicer/usable if you may have your figure for conceptual/theoretical framework earlier at LR section, although later I can see it with results in your Figure 1 in findings section. 

: Figure 1 is now added as you requested (p.6)

  • Your methodology is improved, however I recommend that you may avoid starting your sentence with numbers especially in reporting your data and sample (3.1). You may start with "A total of..." instead. Check spelling Measurement (3.3).

: The wordings have been improved as you requested (p.6 & 7).

  • For analysis (3.2), you may check the latest references to report/write your SEM analyses and findings, and not only referring to Baron & Kenny method and Zhao et al (2010).  I recommend you may refer to Table 1 of H. Kang, J.-W. Ahn (2021) from Asian Nursing Research 15 pp157-162, or any other SEM articles. This is important to make sure that all your results, tables and figures in Section 4 are suffice and relevant to report measurement and structural models.

: Most of the requirements for testing SEM were already done in this study. We acknowledge that there are diverse criteria for testing SEM, and the results we provided may not be sufficient. However, we hope you understand that it would be difficult for us to learn a different way of presenting and interpreting results of SEM with such a short notice (5 days).

  • Perhaps you may consider to add Sobel test or bootstrapping to justify your mediation analysis, since you mentioned one of your contributions is testing causal effect right?

: To be honest, we are not entirely sure what you have in mind. But our best guess is that you are saying that Sobel test or bootstrapping is a better method to test causal effect than SEM. Although I appreciate your suggestion, there also are scholars arguing that SEM is better than Sobel test. Please also note that we are only given five days to revise based on your requests, and that the editor requested minor, rather than major, revision. With all due respects, we will have to ask you to understand that every method has its own weaknesses.

  • I am more familiar with PLS-SEM where even the measurement model will be reported differently than your approach. Therefore, it is worth to check your report writing with the updated articles in reporting SEM.

: We will try to do this when given enough time.

  • Finally, it is also worth if you can justify why you referred and used data of FEVS in the year of 2015, rather than the latest ones? I checked the link and found 2016-2019 and not sure about, 2020-2021 may not available because of pandemic etc? I suspect that is why initially I thought that your study was similar to Fernandez 2015, since you may get access to the latest reports for the recent years, nevertheless you mentioned it was collected in 2015 (which was dated for primary data). Now I possibly understand more about your secondary data, perhaps it is just that you have to clarify why you used 2015.  All the best.

: We redid the analysis using the most recent data which is 2019, and changed the numbers accordingly. As you can see in the texts in red, there was no major difference. Please also note that after covid-19 they changed the survey tools and many of the items we used are no longer available.